# Microbial Phytase in a Diet with Lupine and Extruded Full-Fat Soya Seeds Affects the Performance, Carcass Characteristics, Meat Quality, and Bone Mineralization of Fatteners

**DOI:** 10.3390/ani13101655

**Published:** 2023-05-16

**Authors:** Anna Buzek, Anita Zaworska-Zakrzewska, Małgorzata Muzolf-Panek, Małgorzata Kasprowicz-Potocka

**Affiliations:** 1Department of Animal Nutrition, Faculty of Veterinary Medicine and Animal Science, Poznan University of Life Sciences, Wołyńska 33, 60-637 Poznań, Poland; anna.buzek@up.poznan.pl (A.B.); malgorzata.potocka@up.poznan.pl (M.K.-P.); 2Department of Food Quality and Safety Management, Faculty of Food Science and Nutrition, Poznan University of Life Sciences, Wojska Polskiego 31, 60-637 Poznań, Poland; malgorzata.muzolf-panek@up.poznan.pl

**Keywords:** pigs, phytase, performance, fatty acid profile, meat quality, bone mineralization

## Abstract

**Simple Summary:**

Plants store phosphorus mainly in phytate form, which is slightly digestible by pigs due to the lack of native phytase. Some thermal processes, such as extrusion and exogenous phytase additives, can reduce phytate content in the diet, but the effectivity of the enzyme in diets including processed and unprocessed legume seeds needs to be better recognized. This study aims to investigate the effect of the addition of two phytase dosages to a diet containing raw lupine seeds and full-fat extruded soya seeds on the performance of growing pigs in terms of their carcasses, pork quality, and bone mineralization. Phytase significantly improved the pigs’ growth and feed utilization in the starter period and affected some meat, fat, and bone parameters. Higher phytase dosages did not improve the pigs’ performance, but only increased calcium deposition in their bones.

**Abstract:**

This study aims to determine how different doses of phytase in diets including extruded soya and lupine seeds affect fatteners’ performance, meat quality, bone mineralization, and fatty acid profile. Sixty pigs were divided into three treatment groups. The control group was offered a diet without phytase, whereas the Phy100 and Phy400 groups were provided with 100 g and 400 g of phytase per ton of their diet, respectively. The animals from both experimental groups were characterized by a significantly (*p* < 0.05) higher body weight gain and lower feed efficiency in the starter period than the control group. Unfortunately, their meat had lower (*p* < 0.05) fat content, gluteal muscle thickness, and water-holding capacity. In the meat, a higher phosphorus content (*p* < 0.05) was found, and in the bones, a higher calcium (for Phy400) content was found when phytase was added to the pigs’ diet. The pigs from the Phy100 group tended to have higher mean backfat thickness and C18:2 n-6 content in their fat, but lower C22:5 n-3 content, than the other groups. A higher dosage of phytase is not necessary for the diets of fatteners with extruded full-fat soya and lupin seeds.

## 1. Introduction

Feed is the costliest element of livestock production. Modern farm animal nutrition standards are based on protein-rich components, mainly soybean meal (80% GMO) [1,2]. In May 2020, as part of the Green Deal, the European Commission proposed a “Farm to Fork Strategy” for a “fair, healthy, and environmentally friendly food system”. This strategy aims to achieve a range of goals: cutting the overall use of and risk posed by chemical pesticides by 50%, farming 25% of the EU’s agricultural land organically, reducing nutrient losses (especially nitrogen and phosphorus) by at least half, lowering the use of fertilizers, and halving food waste. The Commission intends to reduce the EU’s dependency on imported feed, such as GMO soybean meal (SBM). It aims to achieve this by promoting plant protein grown in Europe. For the aforementioned reasons, there is a need to focus on other local protein sources which could be safely implemented in pigs’ diets. This would also be in line with the “From Farm to Fork” strategy. This concept implies a series of values for agricultural products: food safety, quality, taste, nutritional and health aspects, diversity, animal welfare, and respect for the environment.

In recent years, increased interest in soybean cultivation has been observed in Europe [3,4,5]. The main product of soya is oil, but expensive equipment is needed to remove the fat from the seeds. On the other hand, full-fat seeds have a high oil content (>20%), which may be beneficial as an energy source in the diets of animals, especially when the prices of energy sources are extremally high. Legumes such as European non-GMO soya and sweet lupine seeds may be essential alternatives to imported SBM [4]. Yellow lupine has the highest protein content among all of the legumes, with an average value of 42% in dry matter (DM), of which more than 85% is digestible [6]. Full-fat soya seeds contain about 25–30% protein and about 20–25% fat in the DM of the seeds [5]. The nutritional value of raw lupin and soya seeds is lower, however, than SBM because of the presence of several anti-nutrient compounds, such as protein inhibitors, lectins, urease, and phytate. The use of legumes in animals’ diets could be improved by thermal processing (i.e., extrusion), which can reduce the amount of thermolabile anti-nutrients. In addition, some enzymes, such as proteases, phytases, and carboxylases, can reduce anti-nutrients and/or improve seed utilization. It is commonly known that the content of phytate is quite high in raw soya seeds (0.3–0.4% DM), lupine seeds (0.40% to 0.9% DM), and all of the grains (0.5–2% DM) [7]. The addition of phytase increases the energy value of the feed, and it can significantly improve its digestibility, including the dry matter, amino acids, total protein, and minerals (e.g., phosphorus (P)) [8,9,10,11]. Moreover, phytase additives can reduce the cost of feeding by reducing phosphates from non-renewable sources in the diet as well as the number of nutrients released into the environment [12]. The use of microbial phytase in feed mixtures with legume seeds has been evaluated in only a few nutritional tests so far, but the current research focuses on examining the effectiveness of the use of phytase in fatteners under typical commercial production conditions [9]. Some studies have shown that phytase in higher doses results in more advantageous growth performances in pigs than when using standard phytase doses (usually levels well over 500 FTU/kg and up to 2500 FTU/kg) [13,14,15,16]. On the other hand, feed mixtures with higher enzyme doses can be more expensive. Based on the literature, it is also clear that the use of high-dose phytase additives in feed mixtures for fatteners with processed sources of vegetable protein has been insufficiently assessed, but it could have beneficial effects on some parameters of carcass quality [13,17,18,19]. This is why this study aimed to determine the effect of adding phytase to diets containing lupine seeds and extruded full-fat soybeans seeds on the growing pigs’ performance, slaughter efficiency, and post-slaughter parameters.

## 2. Materials and Methods

### 2.1. Plant Material

Low alkaloid yellow lupine seeds (*Lupinus luteus* var. Mister) obtained from the Plant Breeding Station in Tulce (Wielkopolska, Poland) were used in the experiment. The soya seeds, cv. Augusta, were obtained from Plant Breeding Stations (HR UP Poznań, Poland).

The seeds were crushed, ground, and heated. The extrusion process was conducted for 10 s at a temperature of 130 °C and a pressure of >20 MPa using a Farmet FE 250 extruder (Ceská Skalice; Czech Republic).

### 2.2. Phytase

The experiment used Phytase Quantum Blue 5G^®^ (AB Vista Feed) produced by *E. coli.* According to the producer’s recommendations, the conditioning of the feed mixture or granulation should not exceed a temperature of 90 °C and should last no longer than 30 s. The minimum phytase activity is 500 FTU/g, and the recommended dose is 100 g of the phytase per ton of the mixture.

### 2.3. Ethical Statement

The experimental procedure complied with the guidelines of the Local Ethical Committee for Experiments on Animals in Poznań regarding animal experimentation and care of the animals under study (European Union (EU) directive 2010/63/EU for animal experiments) [20]. Individual approval for this trial was not required because of the production standards used in this study. The pigs were vaccinated and had unlimited access to feed and water. All of the samples were collected after slaughter.

### 2.4. Animals, Diets, and Experimental Design

Sixty weaners (Polish Large White × (Duroc × Pietrain)) of about 27.6 ± 4.2 kg were divided into 2 groups of 20 animals and kept within the assigned group in 1 pen (10 ♂, 10 ♀). Each pig was marked with an earring with a unique number. The experiment lasted 93 days and was divided into three phases: period 1–24 days, period 2–35 days, and period 3–34 days. Yellow lupine (YL) seeds and full-fat extruded soya seeds (ExS) were the primary protein sources in the feed mixtures given to all groups of animals. The control group (Phy0) was offered a diet without phytase. In group Phy100 and group Phy 400, 100 g and 400 g of phytase per ton of the mixture were added, respectively. The phytate content and native phytase activity in the pigs’ diets were not measured. The complete diets were formulated according to the GfE [21] recommendations, as shown in Table 1.

The health and welfare of the animals were monitored twice a day. After the completion of each fattening phase, the body weight gain (BWG) was individually controlled for each pig. The feed intake was measured for each pen (FI) in each phase and the intake was calculated per group. Based on the results in the FI group, the individual BWG feed conversion ratio (FCR) was calculated. All animals were fed until they reached a final body weight of approximately 120 kg. At the end of the experiment, all of the pigs were transported to a local professional abattoir, stunned by electric shock, and killed by exsanguination. The materials for the analysis were investigated in a total of 12 pigs from each group (average 6 ♂ and 6 ♀).

### 2.5. Analytical Procedures

#### 2.5.1. Feed Analyses

Representative feed samples were taken from the silo and the trough and analyzed in duplicate. The feed material was ground (Retsch Zm 200 ultra-centrifugal mill, Retsch, Haan, Germany) with 1.0 mm sieves and analyzed for crude protein, crude fat, crude fiber, crude ash, and total calcium (Ca) and P according to the AOAC methods 976.05, 920.39, 978.10, 984.27, and 965.17 [22].

#### 2.5.2. Carcasses and Meat Analyses

For the analysis, *m. longissimus thoracis et lumborum* was taken from behind the last thoracic vertebra (T13). From each muscle, in the same place, a 500 g f representative sample was taken, frozen, and homogenized. The carcasses were measured and weighed, and the post-mortem yield (%) was calculated. The meatiness (%) and loin depth (mm) were measured using an IM-03 ultrasound apparatus. Moreover, the linear thickness of the back fat at three points, the thickness of the back fat above the shoulder blade and at the last rib (mm), the thickness of the buttock muscles and the loin, and the length (cm) and width of the carcasses (cm) were measured. The slaughter value of the fattened pigs and the meat quality (nutrient content, color, tenderness, drip loss, cooking loss, and water-holding capacity) were analyzed according to the method described by Lisiak et al. [23].

#### 2.5.3. Meat Fatty Acid Profile

A 3 g sample of the homogenized meat was extracted with 30 mL of Folch solution I (chloroform: methanol = 2:1. *v/v*). Then, the homogenate was filtered with a Whatman No.1 paper filter. The extracts were evaporated to dryness in a nitrogen stream and methylated with a mixture of anhydrous methanol and sulfuric acid (1:5. *v/v*). In the next step, 0.5 mL of methanol was added to the extract containing the lipids and a mixture of 0.15 mL methanol/sulfuric acid (1:5. *v/v*) was added. The samples were heated at 70 °C for 15 min and cooled. Next 0.5 mL of n-hexane was added, followed by the addition of a sufficient amount of water to form two layers. The upper hexane layer was removed and analyzed on a gas chromatograph (Agilent 5890 II-5301 Stevens Creek Blvd. Santa Clara, CA 95051, USA) equipped with a flame ionization detector and fitted with a Supelcowax 10 column (30 m × 0.25 mm I.D., 0.25 mm film thickness). Peaks were identified by comparing the sample peak retention times with those of known methylated fatty acid compounds.

#### 2.5.4. Bone Analysis

From the 12 pigs, in each group, the third metacarpals from the right foot were collected, boiled to remove tissues and cartilage caps, ground, and extracted to remove the fat. Following this, the samples were burned in a muffle furnace (P330, Nabertherm GmbH, Lilienthal, Germany) at 600 °C for 5 h [24]. The P and Ca content were determined according to procedures 984.27 and 965.17 of AOAC [22].

### 2.6. Statistical Analysis

The statistical analysis was performed using SAS 9.3 software (AS Institute, Cary, NC, USA). Normal distribution was tested using the Shapiro–Wilk test and variance homogeneity was tested using the Cochran–Hartley–Bartlett test. The significance levels of differences among the groups in the experiment on the pigs were calculated using one-way ANOVA with Duncan’s test in post hoc analysis (indication of homogenous groups). The obtained results were analyzed statistically by calculating the arithmetic mean and adding the SD (standard deviation). All analyses were calculated at a confidence level of α = 0.05. The correlation between the feed intake (FI) and the addition of phytase was tested using Spearman’s rank correlation coefficient (p. R).

## 3. Results

All pigs remained healthy and readily consumed their feed throughout the study. No lameness, hernias, or other symptoms affecting their well-being were noticed.

### 3.1. Performance Parameters

The results for the performance parameters are shown in Table 2. The animals from both experimental groups offered a diet with phytase additives were characterized by significantly higher body weight gain and lower FCR in the starter period compared to the control group (*p* < 0.05). The enzyme additive did not impact the growing pigs’ performance in the other phases or in the experiment as a whole. A statistically significant correlation was observed between the FCR and the FI. The R Spearman’s correlation coefficient equaled 0.565 (*p* < 0.001) when it was calculated for all obtained data (all groups, namely Phy, Phy100, and Phy400).

### 3.2. Carcass and Meat Quality

The chemical composition of the meat and some parameters of meat quality are presented in Table 3. The meat derived from the pigs fed mixtures including microbial phytase was characterized by significantly lower fat content and water-holding capacity, but higher phosphorus content, than the control group (*p* < 0.05). A tendency toward a higher protein content in the meat in both experimental groups was found (*p* = 0.097). The meat in all of the groups was characterized by similar and proper acidification, color, tenderness, cooking loss, and natural drip loss. There were no significant differences between the experimental groups.

The results for the carcass characteristics were not significantly different for most parameters except for the gluteal muscle thickness, which was significantly lower (*p* = 0.003) in both of the groups receiving phytase in their diet (Table 4).

### 3.3. Fatty Acid Profile

None of the groups differed significantly in the level of fatty acids (*p* > 0.05). There was only a tendency toward an increase in C22:5 n-3 content (*p* = 0.076) in the Phy400 group and an increase in C18:2 n-6 (*p* = 0.098) content in the meat of the Phy100 group (Table 5).

### 3.4. Bone Characteristic

The content of Ca, P, and ash in the pigs’ metacarpal bones is shown in Table 6. The diet with a higher phytase level significantly increased the Ca content as compared to the control group (*p* = 0.025). A tendency toward higher P content in the bones of the pigs from both experimental groups was found (*p* = 0.093), as well as a tendency toward higher ash content in the Phy400 group.

## 4. Discussion

The effect of microbial phytase depends on the dietary phytate concentration, the source of phytate and phytase, and the phytase dosage [8,25]. Raw soybeans contain about 0.4% phytic phosphorus, and some research has shown that its content is reduced to about 0.05–0.06% by extrusion [26]. As previously stated by Zaworska et al. [27] and Zaworska-Zakrzewska et al. [28], extrusion improved the nutritional quality of pea and faba bean seeds, especially through the decrease in the level of anti-nutritional components, including phytate-P content. This is probably due to the fact that during extrusion, some molecules of inositol hexaphosphate may hydrolyze to penta-, tetra-, and triphosphates. In contrast, unprocessed cereal and lupine seeds are quite rich in phytate, containing from 0.5 to 2% phytate in their DM. If these components constitute more than 85% of the feed mixture for pigs, the addition of phytase may be justified. In the current experiment, the pigs’ diets were based on extruded full-fat soya seeds and lupine seeds, and the animals that consumed diets with phytase had significantly higher body weight gain and lower FCR in the starter period. This means that phytase is especially effective in earlier periods, during which animals grow more quickly and have a greater need for nutrients. Moreover, a significant correlation was observed between the FCR and the FI. This is probably connected to improved nutrient digestibility after the decomposition of phytate structures as well as after being freed of sugars and proteins, which could be bonded to the phytate [9,29]. This has been proven in this study through the observation of higher mineral content in the pigs’ metacarpal bones, especially for calcium in the Phy400 group, but also for ash and phosphorus (a tendency for) and higher P deposition in the meat. Correct phosphorus and calcium levels in the diet are necessary for bone mineralization. Interestingly, phytase addition in both groups did not significantly increase the phosphorus content in the bones, which was expected. Other authors have also found that this ability increases with the increase in the enzyme dose [8,16]. Tsai et al. [30] assumed that the P released by phytase is absorbed and contributes to improved bone growth, superior rates of tissue accretion, and increased body weight, but it does not change tissue P concentrations, which was found in the current study. Da Silva et al. [31] noted that phytase treatment did not influence the feed conversion ratio, carcass weight and yield, backfat thickness, loin depth, or lean carcass meat. Some studies have shown that higher phytase dosing results in better growth performance in pigs than standard phytase doses, but this was not proven in this study, which compares both experimental groups. Wiśniewska et al. [10] and Wu et al. [32] found a positive correlation between percentage improvements in feed efficiency in response to phytase and dietary phytate content, including diets based on legume feed components. Grela et al. [33] observed that dietary phytase treatment at 500–1000 FTU per kg significantly influenced daily gains and the FCR, but the FI was unaffected. It should be underlined that there were no significant differences in the production parameters during the entire experimental period. However, pigs from the group with the recommended phytase dose achieved a 4 kg higher total gain than the control group, which could benefit pig farmers. Moreover, in the current study, phytase addition was given “on top”, similarly to the practice of most pig farms. This research could be improved by lowering the phosphate content in the pigs’ diet, which could reduce feed costs slightly. Possibly, in the future, the use of a microbial phytase that hydrolyzes all phosphate bonds in the inositol ring should be considered.

The current study also found that the meat derived from the pigs fed mixtures with microbial phytase was characterized by a significantly lower fat content, gluteal muscle thickness, and water-holding capacity, but the mechanisms of these changes are not clear. Lower fat content in meat, from one side, is attractive for consumers who prefer a low-fat diet, but it could also negatively affect the sensory properties of the meat [18,31]. WHC is the ability of meat to hold all or part of its water, and it is one of the most important traits of meat quality. Weight loss due to drip loss results in higher economic costs for meat processors and retailers. Hollowey et al. [17] found that a higher phytase dose affected pigs’ growth rate, feed intake, and carcass yield (*p* > 0.10), but Gebert et al. [18], who offered pigs 1200 FTU of phytase/kg of feed, found that the slaughter yield, the percentage of lean cuts, and the total fat tissue, as well as the back fat thickness, were not affected by phytase. The results are probably due to the different phytases used in these experiments. The addition of phytase did not significantly impact the pigs’ fatty acids profiles, although the tendency toward an increase in linolenic acid (LA, C18:2 n-6) content in the meat of the animals from the Phy100 group and an increase in docosapentaenoic acid (DPA, C22:5 n-3) in the meat of the pigs from the Phy400 group suggested that phytase could be implicated in some changes in the pigs’ fat structure, but this needs further verification. Considering recent trends, sustainability can be improved with a shift from a dependence on optimally formulated feeds based on imported feed ingredients to the use of feed based on locally sourced legume seeds with additives in pig diets, which can provide opportunities to diversify the feed matrix thanks to homegrown feed ingredients.

## 5. Conclusions

Based on our research, we found that adding phytase in a basic dose may be beneficial in diets containing extruded full-fat soya and lupine seeds, as it enhances the performance of fattening pigs, especially in the starter phase, and positively affects mineral deposition in the pigs’ bones and meat. Unfortunately, phytase negatively affects some carcass and meat parameters, as it reduces the fat content in the meat and its water-holding capacity. In this kind of diet, using a higher dose of phytase seems to be unnecessary.

## Figures and Tables

**Table 1 animals-13-01655-t001:** The composition and content of nutrients in the experimental diets (g/kg of feed).

Components/Group	Period 1			Period 2			Period 3		
	Phy0	Phy100	Phy400	Phy0	Phy100	Phy400	Phy0	Phy100	Phy400
Wheat	250.0	250.0	250.0	-	-	-	-	-	-
Maize	180.6	180.5	180.2	413.3	413.2	412.9	383.3	383.2	382.9
Wheat bran	150.0	150.0	150.0	180.0	180.0	180.0	200.0	200.0	200.0
ExS	100.0	100.0	100.0	75.0	75.0	75.0	50.0	50.0	50.0
Rye	100.0	100.0	100.0	200.0	200.0	200.0	250.0	250.0	250.0
Barley	100.0	100.0	100.0	-	-	-	-	-	-
YL	80.0	80.0	80.0	100.0	100.0	100.0	90.0	90.0	90.0
Limestone	14.5	14.5	14.5	14.0	14.0	14.0	13.0	13.0	13.0
HCL-Lysine 78.5%	5.6	5.6	5.6	4.4	4.4	4.4	4.2	4.2	4.2
Sodium chloride	4.0	4.0	4.0	4.0	4.0	4.0	4.0	4.0	4.0
Liquid acidifier **	3.5	3.5	3.5	-	-	-	-	-	-
1-calcium phosphate	3.0	3.0	3.0	2.5	2.5	2.5	-	-	-
DL-methionine 99%	2.8	2.8	2.8	1.0	1.0	1.0	0.7	0.7	0.7
Premix *	2.0	2.0	2.0	2.0	2.0	2.0	1.5	1.5	1.5
Magnesium oxide	2.0	2.0	2.0	2.0	2.0	2.0	2.0	2.0	2.0
L-threonine 98.5%	1.1	1.1	1.1	1.3	1.3	1.3	0.8	0.8	0.8
L-tryptophan 98%	0.5	0.5	0.5	0.3	0.3	0.3	0.3	0.3	0.3
Choline chloride	0.4	0.4	0.4	0.2	0.2	0.2	0.2	0.2	0.2
Quantum Blue 5G	-	0.1	0.4	-	0.1	0.4	-	0.1	0.4
Nutritional value (g/kg)									
Crude protein	170.7	171.2	168.7	158.9	157.4	158.7	150.4	149.8	150.2
Crude fiber	52.9	52.6	52.1	54.3	54	53.8	54.1	53.8	53.3
Crude fat	43.7	42.9	44.1	45.1	45.9	44.8	40.3	41.2	40.9
P	5.60	5.70	6.70	5.70	5.60	5.60	4.90	5.00	5.00
Ca	7.90	7.80	7.90	7.40	7.30	7.30	6.40	6.30	6.40
Phytase FTU/kg	0	500	2000	0	500	2000	0	500	2000

DM—dry matter; Phy0—control group; Phy100—control group + phytase 100 g/t; Phy400—control group + phytase 400 g/t; YL—yellow lupine, ExS—extruded soya seeds. Premix *—the mineral and vitamin premix contained the following amounts of components per 1 kg: vitamin A—10,833 mg/kg, vitamin D3—25 mg/kg, vitamin E—50,000 mg/kg, vitamin K3—2000 mg/kg, vitamin B1—1000 mg/kg, vitamin B2—2000 mg/kg, vitamin B6—1500 mg/kg, vitamin B12—15 mg/kg, pantothenic acid—1500 mg, nicotinic acid—1000 mg/kg, biotin—50 mg/kg, folic acid—750 mg/kg, Fe—50,000 mg/kg, Mn—37,000 mg/kg, Zn—50,000 mg/kg, Cu—10,000 mg/kg, I—750 mg/kg, and Se—200 mg/kg. Liquid acidifier **—formic acid—500 g/kg, propionic acid—230 g/kg, ammonium formate—200 g/kg, ammonium propionate—40 g/kg, demineralized water—28 g/kg, glycerin—1 g/kg, and glycol—1 g/kg.

**Table 2 animals-13-01655-t002:** The pigs’ performance parameters.

Phase/Variable	Phy0	Phy100	Phy400	SD	*p*-Value
	Period 1				
IBW (kg)	27.66	27.72	27.55	4.21	0.992
FBW (kg)	46.67	49.81	48.92	5.91	0.227
**BWG (kg)**	**19.01 ^b^**	**22.09 ^a^**	**21.37 ^a^**	**2.90**	**0.001**
FI (kg)	48.35	48.33	47.66	-	-
**FCR (kg/kg)**	**2.58 ^a^**	**2.22 ^b^**	**2.27 ^b^**	**0.09**	**0.004**
	Period 2				
FBW (kg)	83.20	86.80	85.00	8.20	0.375
BWG (kg)	36.52	37.04	36.09	0.94	0.756
FI (kg)	98.50	98.65	98.84	-	-
FCR (kg/kg)	2.73	2.69	2.80	0.18	0.546
	Period 3				
FBW (kg)	119.11	123.24	120.68	8.82	0.355
BWG (kg)	35.92	36.4	35.67	3.16	0.765
FI (kg)	122.00	121.25	121.75	-	-
FCR (kg/kg)	3.44	3.35	3.43	0.04	0.636
	Periods 1–3				
BWG (kg)	91.46	95.52	93.13	7.26	0.208
FI (kg)	268.85	268.23	269.26	-	-
FCR (kg/kg)	2.96	2.82	2.93	0.19	0.145

Phy0—control group; Phy100—control group + phytase 100 g/t; Phy400—control group + phytase 400 g/t; IBW—initial body weight; FBW—final body weight; BWG—body weight gain; FI—feed intake; FCR—feed conversion ratio; SD—standard deviation; SEM—standard error of the mean; ^a,b^—means with different superscripts in a row are different *(p* ≤ 0.05).

**Table 3 animals-13-01655-t003:** The chemical composition of the *m. longissimus lumborum* and some parameters of meat quality.

Variable	Phy0	Phy100	Phy400	SD	*p*-Value
Fat (g/kg)	**32.3 ^a^**	**25.9 ^b^**	**25.8 ^b^**	**0.061**	**0.017**
Protein (g/kg)	236.7	242.8	241.2	0.072	0.097
Water (g/kg)	720.0	719.4	721.3	0.075	0.969
**Phosphorus (g/kg)**	**10.7 ^b^**	**11.9 ^a^**	**11.8 ^a^**	**0.080**	**<0.001**
pH 24 min.	5.48	5.50	5.54	0.087	0.197
L*—lightness	49.93	50.13	49.51	1.925	0.681
a* —redness	5.81	5.58	5.29	0.813	0.341
b*—yellowness	3.20	2.88	2.66	0.774	0.334
Cooking loss (%)	28.68	28.12	28.04	4.860	0.954
Tenderness (%)	21.72	20.88	22.78	5.560	0.637
Drip loss (%)	6.54	6.70	5.84	1.685	0.339
Water-holding capacity (%)	**32.18 ^a^**	**28.50 ^b^**	**29.09 ^b^**	**3.330**	**0.011**

Phy0—control group; Phy100—control group + phytase 100 g/t; Phy400—control group + phytase 400 g/t; P—phosphorus; SD—standard deviation; SEM—standard error of the mean; ^a,b^—means with different superscripts in a row are different *(p* ≤ 0.05).

**Table 4 animals-13-01655-t004:** The carcass characteristics of the growing pigs.

Variable	Phy0	Phy100	Phy400	SD	*p*-Value
Carcass weight (kg)	92.95	96.55	94.81	7.90	0.436
Meatiness (%)	56.41	55.46	56.56	3.089	0.483
Cold dressing yield (%)	77.95	78.38	78.68	3.46	0.316
Carcass length (cm)	86.54	86.00	86.31	1.958	0.794
Carcass width (cm)	36.85	38.08	37.15	2.109	0.501
Back (cm)	25.77	23.92	25.15	6.197	0.804
Shoulder (cm)	44.85	41.00	40.46	6.642	0.169
Mean loin depth (mm)	69.14	70.35	71.86	5.592	0.112
Mean backfat thickness (mm)	19.14	20.86	19.40	4.636	0.328
pH	6.68	6.52	6.50	0.239	0.235
Backfat thickness K III (mm)	21.69	19.17	19.38	5.417	0.552
Backfat thickness K II (mm)	16.69	16.33	16.77	3.276	0.374
Backfat thickness K I (mm)	28.00	26.25	25.85	4.337	0.371
**GMT (mm)**	**76.77 ^a^**	**72.67 ^b^**	**71.77 ^b^**	5.683	**0.003**
EC (mS/cm)	3.73	4.24	3.88	1.904	0.688

Phy0—control group; Phy100—control group + phytase 100 g/t; Phy400—control group + phytase 400 g/t; GMT—gluteal muscle thickness; EC—electrical conductivity of muscles; SD—standard deviation; SEM—standard error of the mean; ^a,b^—means with different superscripts in a row are different *(p* ≤ 0.05).

**Table 5 animals-13-01655-t005:** Profile (% of total FA) of fat from the meat.

Variable	Phy0	Phy100	Phy400	SD	*p*-Value
Fatty acids					
C14:0	1.13	1.15	1.18	0.103	0.531
C16:0	22.53	22.12	22.89	1.384	0.429
C17:0	0.22	0.23	0.23	0.018	0.166
C18:0	13.07	13.06	13.16	0.599	0.927
C20:0	0.21	0.21	0.21	0.008	0.499
C16:1	2.56	2.52	2.58	0.140	0.587
C17:1	0.25	0.25	0.25	0.014	0.953
C18:1	41.88	41.14	41.43	0.893	0.129
C20:1	0.99	1.02	1.00	0.058	0.429
C18:3 n-3	1.48	1.51	1.39	0.337	0.712
C20:5 n-3	0.39	0.36	0.36	0.044	0.263
C22:5 n-3	0.42	0.40	0.44	0.041	0.076
C22:6 n-3	0.43	0.43	0.40	0.043	0.196
C18:2 n-6	12.60	13.76	12.61	1.523	0.098
C20:2 n-6	0.34	0.34	0.36	0.032	0.466
C20:3 n-6	0.26	0.26	0.26	0.029	0.953
C20:4 n-6	1.13	1.12	1.15	0.077	0.677
C22:4 n-6	0.12	0.12	0.12	0.014	0.652
SFA	37.16	36.78	37.66	1.477	0.372
MUFA	45.68	44.93	45.25	0.911	0.132
PUFA	17.16	18.30	17.08	1.592	0.112
Σ n-3	2.71	2.70	2.59	0.327	0.651
Σ n-6	14.45	15.60	14.49	1.534	0.114
Σ n-6/Σ n-3	5.40	5.86	5.66	0.920	0.481

Phy0—control group; Phy100—control group + phytase 100 g/t; Phy400—control group + phytase 400 g/t; SFA—saturated fatty acids; MUFA—monounsaturated fatty acids; PUFA—polyunsaturated fatty acids; FA—fatty acids; SD—standard deviation; SEM—standard error of the mean.

**Table 6 animals-13-01655-t006:** The mineral analysis of the metacarpal bones.

Variable	Phy0	Phy100	Phy400	SD	*p*-Value
P (g/kg)	169.6	171.4	171.4	0.022	0.093
Ca (g/kg)	**333.5 ^b^**	**338.9 ^ab^**	**345.5 ^a^**	**1.004**	**0.025**
Crude ash (g/kg)	63.1	62.4	66.8	0.056	0.074

Phy0—control group; Phy100—control group + phytase 100 g/t; Phy400—control group + phytase 400 g/t; P—phosphorus; Ca—calcium; SD—standard deviation; SEM—standard error of the mean; ^a,b^—means with different superscripts in a row are different *(p* ≤ 0.05).

## Data Availability

Data are available from the corresponding authors upon reasonable request.

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
