# Peer review of "Microbial Phytase in a Diet with Lupine and Extruded Full-Fat Soya Seeds Affects the Performance, Carcass Characteristics, Meat Quality, and Bone Mineralization of Fatteners"

_animals, 2023, doi:10.3390/ani13101655_

Round 1

Reviewer 1 Report

The manuscript entitled "Microbial phytase in a diet with lupine and extruded full-fat soya seeds affect performance, carcass characteristics, meat fatty acid profile and bone mineralization of fatteners" is interesting and fits in the content of this journal.

Authors should make some necessary corrections before publishing. With the following corrections, the manuscript may be published in the journal Animals.

Introduction

The authors suggest that higher phytase activity administered in mixtures for fattening pigs will bring better results, but what dictated its high level, which was proposed by the authors of this experiment.

Maybe in the future we should consider using a microbial phytase that hydrolyzes all phosphate bonds in the inositol ring.

The use of higher doses of phytase may increase production costs.

Material and methods

Were animals from one group in the same pen? - please complete it.

Table 1 lacks information on:

phytic phosphorus content and native phytase activity.

If the authors did not analyze the mixtures for these compounds/enzymes - please list them.

In addition, please provide the activity of microbial phytase and express it in FTU/kg and complete Table 1 with this value.

Table 1 - Please explain whether the components were given in % or in g/kg? The table shows that in g/kg. However, the Table says "Components (%)/group".

Table 1 - The authors state "Premix Grower 0.2%*" then where is the composition of the premix for Finisher? - moreover, the description under the table is for "Premix"

Table 1. Please differentiate "*"

If the Authors state "Premix* - The mineral and vitamin premix contained the following amounts of components per 1 kg:" then the content of individual ingredients should be given only in the units "e.g. vitamin K3 - 2000 mg" - the note applies to all ingredients.

Please specify the content of the individual components of "Liquid acidifier* "

How was "The average daily feed intake (ADFI)" controlled since the animals were in one pen - please refer to the manuscript for more details.

Line 128 - please explain on what basis animals were selected for slaughter as part of experimental activities and give the date of slaughter (body weight or day of life) - please complete this in the manuscript.

Line 132 - please complete in the manuscript the date of collection of the mixtures for analysis.

Line 137-144 - were the analyzes done in one repetition?

Line 145 - please complete the manuscript with information on the selection of tissues for meat analysis - What part did the samples come from? How many samples were taken for homogenization? How many partial samples made up one aggregate and later homogenized? How many repetitions were there? e.t.c.

Results

Line 187-188 - please include the correlation data in Table 2. The given correlation concerned which groups? -please explain - make it more specific.

Please don't use p = 0.000, replace it with p< 0.001.

Table 2 - Are the FCR coefficients calculated correctly - please check.

Table 3. Please use SI units (g/kg) – in Table 6 too.

Line 206 and 208 - incorrect pvalue - does not match the data in Table 4.

Line 220 - please remove "Fatty acids" - it is in the title of the Table.

Line 250-251 - please specify the description - Ca increase, in which group in relation to what? - also add pvalue.

Line 357 - error in "e Cola" - verify this

Line 349 please remove "PMID: 35010899; PMCID: PMC8746346."

Line 359 and 397, please adapt the literature to the requirements

Please format your references carefully

Author Response

Dear Reviewer,

We would like to thank You for the very detailed comments to our manuscript, which greatly helped to improve our review. The resubmitted version of the paper was adjusted according to those suggestions. The detailed reply to each of the comments is presented below, including the number of lines where it leads to a change in the paper. We were used "track and changes" function in the revised manuscript. I add the Certificate English editing, no - 65847.  The manuscript in its revised form has been approved by all authors.

Answer below: 

REVIEWER I

  1. Introduction The authors suggest that higher phytase activity administered in mixtures for fattening pigs will bring better results, but what dictated its high level, which was proposed by the authors of this experiment.

That is truth, although our results did not proof it. Many authors found this phenomenon.

  1. Maybe in the future we should consider using a microbial phytase that hydrolyzes all phosphate bonds in the inositol ring.

It is a good idea. We will consider it in the next study in the near future.

  1. The use of higher doses of phytase may increase production costs.

We agree in this suggestion and we add some information in text. L: 90-91

  1. Material and methods, Were animals from one group in the same pen? - please complete it.

             Yes, they were. We added information L: 123

  1. Table 1 lacks information on: phytic phosphorus content and native phytase activity. If the authors did not analyze the mixtures for these compounds/enzymes - please list them.

Phytate content and native phytase activity in diet were not measured. It was added in M&M section. L:.  130-131

  1. In addition, please provide the activity of microbial phytase and express it in FTU/kg and complete Table 1 with this value.

Thank you for your suggestion, the activity of microbial phytase was added under the table 1.

  1. Table 1 - Please explain whether the components were given in %or in g/kg? The table shows that in g/kg. However, the Table says "Components (%)/group".

% was removed. It is in g/kg.

  1. Table 1 - The authors state "Premix Grower 0.2%*" then where is the composition of the premix for Finisher? - moreover, the description under the table is for "Premix".

The premix used was the same for both phases. "grower" has been removed from the table, and it was improved.

  1. Table 1. Please differentiate "*"

I was removed.

  1. If the Authors state "Premix* - The mineral and vitamin premix contained the following amounts of components per 1 kg:" then the content of individual ingredients should be given only in the units "e.g. vitamin K3 - 2000 mg" - the note applies to all ingredients.

It was improved.

  1. Please specify the content of the individual components of "Liquid acidifier* "

The specify the content of the components of "Liquid acidifier*  were added in text:

“Liquid acidifier** – formic acid -500g/kg, propionic acid -230g/kg, ammonium formate-200g/kg, ammonium propionate- 40g/kg, demineralized water- 28g/kg, glycerine- 1g/kg, glycol – 1g/kg

  1. How was "The average daily feed intake (ADFI)" controlled since the animals were in one pen - please refer to the manuscript for more details.

ADFI was measured in groups. It was added L: 147

  1. Line 128 - please explain on what basis animals were selected for slaughter as part of experimental activities and give the date of slaughter (body weight or day of life) - please complete this in the manuscript.

All information was added in the manuscript. L: 151-154

  1. Line 132 - please complete in the manuscript the date of collection of the mixtures for analysis.

It was added.

  1. Line 137-144 - were the analyzes done in one repetition?

No, the analyzes done in duplicate – it was added in manuscript L. 158-159

  1. Line 145 - please complete the manuscript with information on the selection of tissues for meat analysis - What part did the samples come from? How many samples were taken for homogenization? How many partial samples made up one aggregate and later homogenized? How many repetitions were there? t.c.

It was added in section M&M. L: 164-166.

  1. Results

Line 187-188 - please include the correlation data in Table 2. The given correlation concerned which groups? -please explain - make it more specific.

  1. We have discussed this point and think, that the correlation information is sufficient to be included in the text. We have added the necessary, detailing information. L:223-224.

  1. Please don't use p = 0.000, replace it with p< 0.001.

It was improved.

  1. Table 2 - Are the FCR coefficients calculated correctly - please check.

Yes, they are ok.

  1. Table 3. Please use SI units (g/kg) – in Table 6 too.

It was improved.

  1. Line 206 and 208 - incorrect pvalue - does not match the data in Table 4.

It was improved.

  1. Line 220 - please remove "Fatty acids" - it is in the title of the Table.

It was removed.

  1. Line 250-251 - please specify the description - Ca increase, in which group in relation to what? - also add pvalue.

It was added. L: 293-294

  1. Line 357 - error in "e Cola" - verify this

It was improved.

  1. Line 349 please remove "PMID: 35010899; PMCID: PMC8746346."

It was done.

  1. Line 359 and 397, please adapt the literature to the requirements

It was improved.

  1. Please format your references carefully

It was checked.

We hope the changes made increased the quality of the manuscript. We would be glad if the manuscript in its revised version could be published in MDPI Animals.

Sincerely,

Anita Zaworska-Zakrzewska

Reviewer 2 Report

Review of the manuscript entitled “Microbial phytase in a diet with lupine and extruded full-fat soya seeds affects performance, carcass characteristics, meat fatty acid profile and bone mineralization of fatteners”.

General comment

The use of phytase in swine nutrition is presently a standard if diets are based on cereal grains. Its positive nutritional effect in such a condition is well known, especially because of improved digestibility, and that is why the usage of phytase is recommended by European Commission as one of important pro ecological action in agriculture. It is, however, poorly known the effect of phytase usage when the main source of nutritional protein is changed. That is why the manuscript describes interesting and practically important experiment, especially because of the strong European trend to reduce importance of imported soybean meal, and replace it with local sources like lupin seeds or full-fat soya seeds from European cultivation. Unfortunately, the manuscript is written relatively poor English language, with lots of grammar mistakes, and must be improved substantially before acceptance.

Specific comments

Title

Title of the manuscript is long, but generally informative, however, in my opinion it sometimes focus on less important effect omitting much better seen. Results show no significant effect of the use of phytase on fatty acid profile, but some effects are visible on general quality traits. That is why I propose to change it to a little more general:

“Microbial phytase in a diet with lupine and extruded full-fat soya seeds affects performance, carcass characteristics, meat quality and bone mineralization of fatteners”

Introduction

This section is built using excessively too long sentences, and that is why sometimes could be difficult to understand, and sometimes they are just improper from grammar point of view. It seems to be that the Authors have written the first text in other language and then made some mistakes during translation, because of differences in grammar rules between languages. In my opinion, the Authors should consult the text with English native speaker. Sometimes the Authors also lose their train of thought, probably because of too long sentences that are poorly connected one to another. Some examples:

The sentence in lines 52-54 focus on cultivation problems, and the next sentence in lines 54-56 focus on fat content in seeds problem and necessity to remove it, but starts from “It means”. There is no connection between cultivation problems and fat content that is why such sentences construction is not justified. The next sentence in lines 56-57 is even more illogical, because starting from “Moreover” focus on positive effect of high fat content. So, we have to sentences describing problems and “moreover” positive effect. This sentence should start from e.g. “on the other hand”, or something like that.

Lines 62-65. Too long sentence, and thus illogical content. The simple meaning of the sentence is that extrusion can reduce not only antinutrients but also enzymes. Enzymes should be described by new independent sentence to omit such doubts.   

Line 70. It is rather impossible to remove phosphates. They can be reduced, but not removed.

Line 73. What does it mean “this paper”. The Authors mean their paper or the paper being cited? It is not clear in this sentence.

Those sentences are only examples, and all the Introduction needs substantial language review to be more clear and informative, and also proper in grammar rules.

M&M

Similarly to Introduction M&M also should be consulted with native speaker. The section should start from characterizing cultivars and origin of both protein sources. The processing of soya seeds should be described separately in additional sentence.

Line 96. Last NO longer than 30 s.

Line 104. Pigs were vaccinated during experiment or before? Against which diseases the were vaccinated? Was vaccination important for methodology or results of experiment? Vaccination should be described much more detailed, or omitted if was not important for methodology or results.

Line 107. Initial body weight should be described as mean +/- SD, not about value. Generally piglets are defined as pigs from birth to wean. 27,6 kg is not piglet but weaner.

Table 1. Is nutritional value calculated for fresh or dry matter?

128-129. Animals were killed in the place of experiment or transported to professional abattoir? This should be clarified.

Line 132-135. What about antinutrients? In the Introduction the Authors focused on antinutrients, and potential effect of enzymes on antinutrients. This seem to be valuable to analyze most important antinutrients that can be affected by phytase.

Line 139. Lard seem to be not good word in this case.   

Lines 139-141. There is no verb in this sentence, and it seems to be incomplete.

Line 165. Improper word order in this sentence

Line 173. Duncan’s test is post hoc, continuing ANOVA. There is no word about ANOVA in M&M.

Line 175. Why Authors decided to use two different dispersion measures (SD and SEM)? It is not typical, and should be described and justified.

Results.

There is also some grammar mistakes in this section.

Lines 183-185. This sentence seem to suggest that pigs offered phytase in diet. This must be amended.

Table 2. There are some doubts concerning data in this table.

a.       FI is not defined as a parameter under table (I can guess that it is feed intake, but it shoul be defined).

b.       FI was calculated per group, but FCR was calculated per pig. How it is possible, that without individual feed intake the Authors calculated individual FCR? This should be described in M&M.

c.       Starting from FCR in Period 2, SD and SEM in every parameter is identical. It seems to be impossible. The Authors should analyze data in the table once more.

Line 194. The term muscle meat seem to be improper. Is there any other meat than muscle?

Table 4. Why Authors did not place loin depth in the table?

Lines 214-216. In M&M the Authors defined that significant difference is in p < 0.05. That is why the first sentence is not truth. The Authors must be more careful in data description. There is no significant difference in fatty acid profile.

Discussion

This section seem to be the best written of all the manuscript. The most important data were discussed taking into consideration results of the other authors. The language in this section also seem to be the most proper. There are only some small mistakes, like grammar of the sentence in lines 285-287.

Conclusions are based on data obtained during experiment, and the Authors clearly underline, that higher dosage of phytase does not improve production traits.

Lines 315-316. Should be “seems TO BE unnecessary”.

To conclude, in my opinion the manuscript describes simple, but interesting experiment with data being important for practice. That is why it is worthy to publish, but it needs substantial improvement before final decision. I suggest minor revision, because of relativel low number of methological concerns adn data presentation mistakes. But I strongly recommend deep language revision by English native speaker.

Unfortunately, the manuscript is written relatively poor English language, with lots of grammar mistakes, and must be improved substantially before acceptance. I strongly recommend deep language revision by English native speaker.

Author Response

Dear Reviewer,

We would like to thank You for the very detailed comments to our manuscript, which greatly helped to improve our review. The resubmitted version of the paper was adjusted according to those suggestions. The detailed reply to each of the comments is presented below, including the number of lines where it leads to a change in the paper. We were used "track and changes" function in the revised manuscript. I add the Certificate English editing, no - 65847.  The manuscript in its revised form has been approved by all authors.

Answer below: 

  1. Unfortunately, the manuscript is written relatively poor English language, with lots of grammar mistakes, and must be improved substantially before acceptance.

English was improved by MDPI English Editing Department – by native speaker Rachel Dunster. We sent to editor certificate no 65847.

Title

Title of the manuscript is long, but generally informative, however, in my opinion it sometimes focus on less important effect omitting much better seen. Results show no significant effect of the use of phytase on fatty acid profile, but some effects are visible on general quality traits. That is why I propose to change it to a little more general:

“Microbial phytase in a diet with lupine and extruded full-fat soya seeds affects performance, carcass characteristics, meat quality and bone mineralization of fatteners”

Thank you for your suggestion, we agree. It was changed.

  1. This section is built using excessively too long sentences, and that is why sometimes could be difficult to understand, and sometimes they are just improper from grammar point of view. It seems to be that the Authors have written the first text in other language and then made some mistakes during translation, because of differences in grammar rules between languages. In my opinion, the Authors should consult the text with English native speaker. Sometimes the Authors also lose their train of thought, probably because of too long sentences that are poorly connected one to another. Some examples:

It was improved by native-speaker.

  1. The sentence in lines 52-54 focus on cultivation problems, and the next sentence in lines 54-56 focus on fat content in seeds problem and necessity to remove it, but starts from “It means”. There is no connection between cultivation problems and fat content that is why such sentences construction is not justified. The next sentence in lines 56-57 is even more illogical, because starting from “Moreover” focus on positive effect of high fat content. So, we have to sentences describing problems and “moreover” positive effect. This sentence should start from e.g. “on the other hand”, or something like that.

Thank you for your suggestion and for capturing this, it was improved.

  1. Lines 62-65. Too long sentence, and thus illogical content. The simple meaning of the sentence is that extrusion can reduce not only antinutrients but also enzymes. Enzymes should be described by new independent sentence to omit such doubts.   

It was changed.

  1. Line 70. It is rather impossible to remove phosphates. They can be reduced, but not removed.

Yes, it is true, it was changed.

  1. Line 73. What does it mean “this paper”. The Authors mean their paper or the paper being cited? It is not clear in this sentence. Those sentences are only examples, and all the Introduction needs substantial language review to be more clear and informative, and also proper in grammar rules.

It was checked and improved.

  1. The section should start from characterizing cultivars and origin of both protein sources. The processing of soya seeds should be described separately in additional sentence.

It was improved.

  1. Line 96. Last NO longer than 30 s.

It was added.

  1. Line 104. Pigs were vaccinated during experiment or before? Against which diseases the were vaccinated? Was vaccination important for methodology or results of experiment?

In the Ethical statement section is information: ….The pigs were vaccinated and had unlimited access to feed and water. All samples were collected after slaughter…. Vaccination details was omitted because it was not important for methodology and results.

  1. Line 107. Initial body weight should be described as mean +/- SD, not about value. Generally piglets are defined as pigs from birth to wean. 27,6 kg is not piglet but weaner.

The level of SD was in table 2, but we added in M&M. L: 122

  1. Table 1. Is nutritional value calculated for fresh or dry matter?

It was added.

  1. 128-129. Animals were killed in the place of experiment or transported to professional abattoir? This should be clarified.

           They were transported to professional abattoir. It was added. L: 151-153

  1. Line 132-135. What about antinutrients? In the Introduction the Authors focused on antinutrients, and potential effect of enzymes on antinutrients. This seem to be valuable to analyze most important antinutrients that can be affected by phytase.

We did not analyse antinutrients in diets.

  1. Line 139. Lard seem to be not good word in this case.

It was changed to back fat L:169

  1. Lines 139-141. There is no verb in this sentence, and it seems to be incomplete.

It was improved.

  1. Line 165. Improper word order in this sentence

It was improved by native-speaker.

  1. Line 173. Duncan’s test is post hoc, continuing ANOVA. There is no word about ANOVA in M&M.

The information has been clearly recorded and improved. L:204-206.

  1. Line 175. Why Authors decided to use two different dispersion measures (SD and SEM)? It is not typical, and should be described and justified.

Thank for your question. We know that, the Standard deviation (SD) measures the amount of variability, or dispersion, from the individual data values to the mean. The standard error of the mean (SEM) measures how much discrepancy is likely in a sample’s mean compared with the population mean. The SEM takes the SD and divides it by the square root of the sample size. We discussed this point and decided to keep only the SD value.

  1. Lines 183-185. This sentence seem to suggest that pigs offered phytase in diet. This must be amended.

It was improved.

  1. Table 2. There are some doubts concerning data in this table. FI is not defined as a parameter under table (I can guess that it is feed intake, but it shoul be defined).

It was done.

  1. FI was calculated per group, but FCR was calculated per pig. How it is possible, that without individual feed intake the Authors calculated individual FCR? This should be described in M&M.

Yes, we calculated individual FCR. Due to the group housing of the animals in each phase ADFI were estimated in the whole group. Based on the possessed ADFI in group and individual BWG feed conversion ratio (FCR) was calculated.

We described in section M&M. L: 147-150

In earlier published studies, we calculated the FCR in the same way. oi: 10.17265/2161-6256/2020.04.000 – experiment I.

  1. Starting from FCR in Period 2, SD and SEM in every parameter is identical. It seems to be impossible. The Authors should analyze data in the table once more.

Thank you for your capturing those errors. We checked all data and changed In the table.

  1. Line 194. The term muscle meat seem to be improper. Is there any other meat than muscle?

It was improved.

  1. Table 4. Why Authors did not place loin depth in the table?

In the table 4 we shown mean thickness of pork loin (mm). We changed the name - mean thickness of pork loin to mean loin depth.

  1. Lines 214-216. In M&M the Authors defined that significant difference is in p < 0.05. That is why the first sentence is not truth. The Authors must be more careful in data description. There is no significant difference in fatty acid profile.

It was corrected.

  1. This section seem to be the best written of all the manuscript. The most important data were discussed taking into consideration results of the other authors. The language in this section also seem to be the most proper. There are only some small mistakes, like grammar of the sentence in lines 285-287.

It was improved by native speaker.

  1. Conclusions are based on data obtained during experiment, and the Authors clearly underline, that higher dosage of phytase does not improve production traits. Lines 315-316. Should be “seems TO BE unnecessary”.

It was improved.

It was checked.

We hope the changes made increased the quality of the manuscript. We would be glad if the manuscript in its revised version could be published in MDPI Animals.

Sincerely,

Anita Zaworska-Zakrzewska

Reviewer 3 Report

Report on the manuscript animals-2380234 entitled: Microbial phytase in a diet with lupine and extruded full-fat soya seeds affect performance, carcass characteristics, meat fatty acid profile and bone mineralization of fatteners.

Showed results:

Table 2: higher BWG but only in the starter phase and then it is observed that it has no impact on the final BW. Therefore, what implications? Useless?

Table 3: lower IMF which could have negative effects on sensory properties, and lower WHC which is something very negative for the industry and the consumer.

Table 4: higher backfat thickness? That should imply lower lean meat yield and carcass performance. I cannot understand why lean meat yield was not affected nor was the IMF content.

Nevertheless, the authors declare:

“We found that adding phytase in the basic dose may be beneficial in diets containing extruded full-fat soya and lupine seeds, as it enhances the performance of fattening pigs, especially in the starter phase, and impacts positively some carcass and meat parameters, and has a tendency to mineral deposition in bones”.

I cannot find such a “positive impact” anywhere.

The authors must rewrite their manuscript from a more realistic point of view.

-          Results must be described in more detail.

-          Discussion must be improved. Critical comments and comparisons with results from other researchers must be included.

-          L. 138 “meatiness” and Table 4 “meatness”. What do the authors mean? Could it be “lean meat yield”? How was it calculated?

-          L. 148-154. Could the authors explain the usefulness of this step?

Extensive editing of the English language required

Author Response

Dear Reviewer,

We would like to thank You for the very detailed comments to our manuscript, which greatly helped to improve our review. The resubmitted version of the paper was adjusted according to those suggestions. The detailed reply to each of the comments is presented below, including the number of lines where it leads to a change in the paper. We were used "track and changes" function in the revised manuscript. I add the Certificate English editing, no - 65847.The manuscript in its revised form has been approved by all authors.

Answer below:

  1. Table 2: higher BWG but only in the starter phase and then it is observed that it has no impact on the final BW. Therefore, what implications? Useless?

It was added. L.: 315-317

  1. Table 3: lower IMF which could have negative effects on sensory properties, and lower WHC which is something very negative for the industry and the consumer.

Thank you for your suggestion.

Lower fat content in meat from one side is attractive for these consumers who prefer low-fat diet, but it also could negatively affect sensory properties of meat.

Thank you for your comments. It was added in discussion. L:.351-355

  1. Table 4: higher backfat thickness? That should imply lower lean meat yield and carcass performance. I cannot understand why lean meat yield was not affected nor was the IMF content. Nevertheless, the authors declare: “We found that adding phytase in the basic dose may be beneficial in diets containing extruded full-fat soya and lupine seeds, as it enhances the performance of fattening pigs, especially in the starter phase, and impacts positively some carcass and meat parameters, and has a tendency to mineral deposition in bones”. I cannot find such a “positive impact” anywhere.

We carefully re-analysed the results and output reports from the statistical program. We found an error in the p-value notation. When writing the data, one "0" was added unnecessarily. The values do not differ significantly from each other, only numerically. We changed the data  in table, and text in section results. In the results we see lower level of meatiness in the Phy100 group ( but it not significant). The conclusion also was improved.

  1. The authors must rewrite their manuscript from a more realistic point of view. Results must be described in more detail. Discussion must be improved. Critical comments and comparisons with results from other researchers must be included.

It was improved.

  1. -          L. 138 “meatiness” and Table 4 “meatness”. What do the authors mean? Could it be “lean meat yield”? How was it calculated?

The meatiness was analysed in local professional slaughterhouse by qualified and experienced personnel  the regression equations for the estimation of this parameter using classification device IM-03. This information is in the text. Line 168.

  1. 6. 148-154. Could the authors explain the usefulness of this step?

The analysis of the meat fatty acid profile was commissioned to a cooperating laboratory, from where the full methodology of the analyzes was obtained. To fully illustrate each step of the procedure, the entire methodology is included in the manuscript. The indicated information is not obligatory. It’s removed in the manuscript.

  1. Comments on the Quality of English Language Extensive editing of the English language required.na

English was improved by MDPI English Editing Department – by native speaker Rachel  Dunster. We sent to editor certificate no 65847.

We hope the changes made increased the quality of the manuscript. We would be glad if the manuscript in its revised version could be published in MDPI Animals.

Sincerely,

Anita Zaworska-Zakrzewska

Round 2

Reviewer 3 Report

The manuscript has improved considerably.

I think that the authors should double-check the changes that the native speaker made because some of them are not correct regarding the scientific context.

For example, the word “approx.” Or “approximately” have been included. It is not appropriate to describe a quantity as “approx.”.

L. 119-120. “of fresh DM”. What does that mean? What is “fresh DM”?

And so on.

--

Author Response

Dear Reviewer,

In the first place we would like to express our thanks for Reviewer next suggestions and comments. 
Fresh DM- means - dry matter in feed. We corrected this in the table.
We tried to correct the manuscript based on reviewer´s demands and added or explain some terms.

We hope the changes made increased the quality of the manuscript. We would be glad if the manuscript in its revised version could be published in MDPI Animals.

Sincerely,

Anita Zaworska-Zakrzewska